# Reasoning Models Better Express Their Confidence

**Dongkeun Yoon**[1]* **Seungone Kim**[3] **Sohee Yang**[4] **Sunkyoung Kim**[2]

**Soyeon Kim**[2] **Yongil Kim**[2] **Eunbi Choi**[2] **Yireun Kim**[2] **Minjoon Seo**[1]

[1]KAIST    [2]LG AI Research    [3]CMU    [4]UCL

dkyoon@kaist.ac.kr

## Abstract

Despite their strengths, large language models (LLMs) often fail to communicate their confidence accurately, making it difficult to assess when they might be wrong and limiting their reliability. In this work, we demonstrate that reasoning models that engage in extended chain-of-thought (CoT) reasoning exhibit superior performance not only in problem-solving but also in accurately expressing their confidence. Specifically, we benchmark six reasoning models across six datasets and find that they achieve strictly better confidence calibration than their non-reasoning counterparts in 33 out of the 36 settings. Our detailed analysis reveals that these gains in calibration stem from the slow thinking behaviors of reasoning models (e.g., exploring alternative approaches and backtracking) which enable them to adjust their confidence *dynamically* throughout their CoT, making it progressively more accurate. In particular, we find that reasoning models become increasingly better calibrated as their CoT unfolds, a trend not observed in non-reasoning models. Moreover, removing slow thinking behaviors from the CoT leads to a significant drop in calibration. Lastly, we show that non-reasoning models also demonstrate enhanced calibration when simply guided to slow think via in-context learning, fully isolating slow thinking as the source of the calibration gains.[2]

## 1 Introduction

A persistent weakness of large language models (LLMs) is their tendency to sound confident even when they are wrong [13, 45]. This overconfidence threatens their reliability, especially in high-stakes scenarios [32, 49]. Meanwhile, recent reasoning models like OpenAI's o1 [30] and Deepseek-R1 [4] have demonstrated strong problem-solving capabilities through chain-of-thought (CoT) [43] eliciting *slow thinking* behaviors such as exploring alternative approaches and verifying their answers [24, 6]. Yet, it remains underexplored whether such slow thinking behaviors also help LLMs to "know what they know" [16] and accurately communicate the limits of their knowledge.

To this end, we present an extensive study of reasoning models' ability to accurately estimate and express confidence in their output; a process known as verbalized confidence estimation [40, 7]. We demonstrate that reasoning models exhibit superior confidence calibration compared to non-reasoning models, and that this improvement stems from slow thinking behaviors, which allow LLMs to dynamically adjust their confidence throughout the reasoning process. For instance, as shown in Figure 1, the model's confidence increases when it first verifies its answer ("Oh, and he also created

---

*Work done during internship at LG AI Research.

[2]Our code is available at https://github.com/MattYoon/reasoning-models-confidence

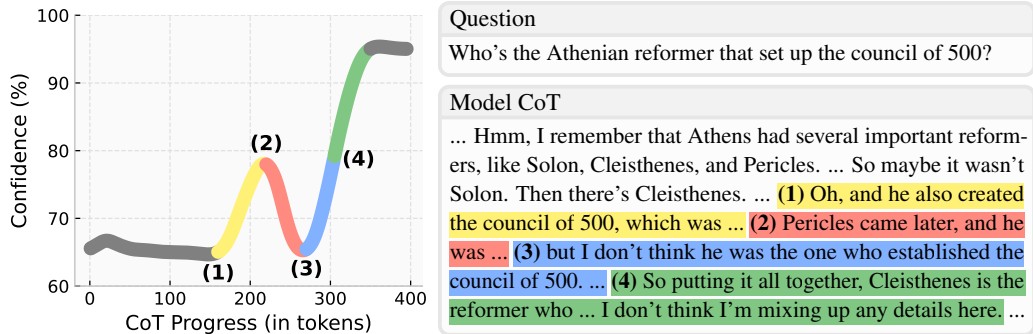

Figure 1: R1-Distill-Qwen-32B dynamically refines its confidence throughout CoT (**left**) as it engages in various slow thinking behaviors (**right**). We collect the model's answer and confidence at each token position by appending "</think>\nAnswer:" to terminate the reasoning process early. Note that the model's answer is consistent and correct ("Cleisthenes") at all points, while the confidence fluctuates. For visual clarity, the data was smoothed using a Butterworth low-pass filter. See Appendix A.4 for the full untruncated CoT.

the council of 500"), and decreases when it considers an alternative ("Pericles came later"). The confidence then rises again as it rejects the alternative ("but I don't think he was"), and reaches its highest level with further verification ("I don't think I'm mixing up any details").

We begin by benchmarking six reasoning models derived from four backbone LLMs against their non-reasoning, instruction-tuned counterparts across two knowledge-focused datasets (TriviaQA and NonambigQA) [15, 23, 17] and four reasoning-intensive datasets (subsets of SuperGPQA and MMLU-Pro) [42, 21]. In 33 out of 36 settings, reasoning models strictly outperform their non-reasoning counterparts across all measured calibration metrics. Notably, reasoning models achieve better calibration even on knowledge-focused datasets, despite having comparable task accuracy to non-reasoning models. This highlights that the calibration gains are not simply derived from superior task performance.

We then conduct a detailed analysis to support our claim that the calibration gains in reasoning models stem from slow thinking, which allows them to dynamically adjust and correct their confidence throughout the reasoning process. (1) We measure changes in calibration metrics throughout the CoT process and observe a *steady, gradual* improvement for reasoning models, with statistically significant trends ($p < 0.05$). Non-reasoning models show no significant trend and in some cases, surprisingly, even exhibit a worsening pattern. (2) In our ablation study, we find that the non-linear structure of slow thinking [6], such as the model's ability to explore alternatives and revise its reasoning, plays a critical role in improving calibration. Conversely, prompting the model to explicitly reason about its own confidence yields only limited gains in calibration. (3) Lastly, we observe that non-reasoning models can also benefit from slow thinking when simply guided via in-context learning [5], supporting our claim that these improvements arise from the slow thinking process itself, rather than from some other factors inherent to reasoning models.

## 2    Related Work

**Confidence in LLMs**    *Confidence* refers to the model's estimated probability that its answer is correct, while *calibration* measures how well this confidence aligns with actual correctness [8]. In other words, a well-calibrated model's confidence is a reliable indicator of prediction correctness [27]. Therefore, assessing an LLM's confidence and enhancing calibration enables more reliable and trustworthy models [7], supporting a wide range of applications such as hallucination detection [22, 41], ambiguity detection [12], uncertainty-guided data retrieval [14], and uncertainty-aware LLM agents [10].

Various approaches have been proposed for *confidence estimation*, which is the process of inferring an LLM's confidence in its predictions [7]. This includes measuring token-level probabilities of answers [9], training proxy probe models on internal hidden states [16], and applying supervised fine-tuning with labeled confidence data [46]. Nonetheless, these methods assume access to a model's

internal states or weights, which are usually unavailable in proprietary LLMs [36, 44]. Additionally, they are often model-specific, necessitating readjustment or retraining when the underlying LLM changes [47]. An alternative approach that avoids these limitations involves sampling multiple responses for the same prompt and deriving confidence based on response consistency [22]. While this method is compatible with proprietary LLMs and is model-agnostic, its primary limitation is the increased computational cost due to repeated inference [47].

In this work, we focus on *verbalized* confidence estimation, where the LLM is prompted to directly express its confidence as a part of their output, either through a linguistic phrase (e.g., "Highly likely") or a numerical probability (e.g., "0.95") [7]. Verbalized confidence estimation has emerged as a widely studied approach as it is compatible with proprietary LLMs, model-agnostic, and computationally efficient [40, 45, 47]. While verbalized confidence estimation offers ease of use and broad applicability, its accessibility also makes its limitations more consequential. Overconfidence is a persistent issue: LLMs tend to express high certainty even when they are incorrect, posing serious risks for real-world deployment of LLMs [45, 50]. Our results suggest that reasoning models and their use of slow thinking hold strong potential to mitigate this issue.

**Reasoning Models**  Increasing compute at inference time, or inference-time scaling, has become a focus for improving the reasoning performance of LLMs [37]. One effective approach to inference-time scaling is training models to generate longer CoTs, as demonstrated by models like OpenAI's o1 [30] and DeepSeek-R1 [4], which are commonly referred to as reasoning models. The key factor that distinguishes the CoT of reasoning models from that of non-reasoning, instruction-tuned models beyond just length is their *slow thinking* [24], which drives their problem-solving ability. Slow thinking mirrors human cognitive behaviors, characterized by non-linear reasoning traces such as exploring alternative approaches, verifying answers, and backtracking [6]. During slow thinking, reasoning models also frequently produce epistemic markers that signal uncertainty (e.g., "I think", "maybe") which are rarely observed in non-reasoning models [50]. Structurally, reasoning models typically enclose their CoT within special tokens such as `<think>...</think>` to isolate the reasoning phase, in contrast to non-reasoning models.

In this paper, we investigate whether reasoning models can more accurately express their confidence by leveraging their slow thinking behaviors. A concurrent study also examines the calibration of reasoning models [48], but focuses on training an external probe over hidden states to optimize CoT generation. In contrast, our study centers on verbalized confidence estimation and provides extensive analysis to uncover how reasoning models are able to better express their confidence.

# 3 Reasoning models better express their confidence

In this section, we demonstrate that reasoning models express their confidence more accurately than non-reasoning models, first detailing the experimental setup (Section 3.1) and then presenting the results (Section 3.2). Full results and additional experiments are provided in Appendix A, and further setup details are provided in Appendix B.

## 3.1 Experiment setup

**Datasets**  We evaluate LLM calibration across two types of datasets: *knowledge-focused* and *reasoning-intensive*. We use the knowledge-focused datasets, TriviaQA and NonambigQA [15, 23, 17], to provide a more controlled setting for comparing the calibration of reasoning and non-reasoning models, as both achieve similar accuracy given that CoT does little to help solve these tasks [38]. Therefore, this setup ensures that any differences in calibration are not simply due to reasoning models being better at solving the task. In addition, since reasoning models are primarily intended for complex reasoning tasks, we also include reasoning-intensive datasets, MMLU-Pro and SuperGPQA [42, 21], to ensure our findings generalize. These datasets are challenging multiple-choice benchmarks centered on knowledge-driven reasoning. We use two subsets of each reasoning dataset: a *Math* subset focused on arithmetic reasoning, and a *Non-Math* subset covering other types of reasoning. Due to our broad range of experiments, we uniformly sample 1,000 examples from each dataset or subset to keep the compute manageable. To assess the variability of this choice, we perform bootstrapping over multiple resampled subsets and report the standard deviation in Appendix A.2.4.

**Models** We benchmark a diverse set of six reasoning models derived from four different 32B-scale LLMs against their non-reasoning counterparts.[3] Specifically, we evaluate: R1-Distill-Qwen, QwQ, and OR1-Preview reasoning models [4, 35, 11] against Qwen2.5-Instruct [33]; GLM-Z1-0414 against GLM-4-0414 [39]; EXAONE-Deep [20] against EXAONE-3.5-Instruct [19]; and Qwen3 with Thinking Mode (abbreviated as Qwen3 Thinking) against Qwen3 with Non-thinking Mode (Qwen3 Non-thinking) [34]. Unlike the other pairs, which consist of separate model checkpoints, Qwen3 is a hybrid model that supports both reasoning-style CoT (Thinking Mode) and non-reasoning-style CoT (Non-Thinking Mode) via prompting.[4]

**Inference procedure** In a single turn of conversation, we instruct the models to perform three steps using CoT: (1) SOLUTION REASONING, where the model reasons step-by-step to arrive at an answer to the given question; (2) CONFIDENCE REASONING, where it evaluates its own confidence for that answer step-by-step; and (3) CONFIDENCE VERBALIZATION, where it maps their confidence in one of ten bins, ranging from "Almost no chance (0–0.1)" to "Almost certain (0.9–1.0)". Each bin includes both a linguistic descriptor (e.g., "Almost certain") and its corresponding numerical probability (e.g., "0.9–1.0"), which are inspired by the approaches used in prior work [40, 45]. To ensure a fair comparison, we use the same instructions for both reasoning and non-reasoning models. The full prompt we use is included in Appendix B.2.

For reasoning models, we expect all three steps to be carried out within the thinking process `<think>...</think>`. However, we observe that some reasoning models (specifically R1-Distill, OR1-Preview, and GLM-Z1) rarely engage in CONFIDENCE REASONING.[5] To address this, we force these models to include CONFIDENCE REASONING within their thinking process by generating up to the `</think>` token, and replacing the token with "Okay, now let's assess my overall thinking process so far step-by-step. I need to evaluate how likely my answer is correct." and performing a second round of inference. We ablate the effect of this choice in Section 4.2, where we observe that it has negligible impact on calibration.

As part of the instruction, models are asked to format their final response as "`Answer:ANSWER Confidence:CONFIDENCE`" after completing all three steps. We apply rule-based filtering using regular expressions to extract the predicted answer and confidence. Answers are matched against the ground truth using GPT-4o mini [28], leveraging the prompt from OpenAI's `simple-evals` codebase [31]. We describe the full procedure we use to ensure that all model responses contain extractable answers and confidence predictions in Appendix B.2.

**Evaluation metrics** We use commonly adopted calibration metrics from prior work [7, 40, 45]. Expected Calibration Error (**ECE**) [26] measures the average discrepancy between accuracy and predicted confidence within each confidence bin, weighted by the number of samples in each bin. While ECE is an intuitive metric for measuring absolute calibration, it fails to capture calibration at the individual prediction level and does not account for the model's discriminative ability, which refers to how well the model assigns higher confidence to correct predictions over incorrect ones [7]. In contrast, **AUROC** [1] captures the model's discriminative ability by computing the probability that a randomly chosen correct prediction is assigned higher confidence than a randomly chosen incorrect one, but it does not measure absolute calibration. **Brier Score** [2] quantifies the mean squared difference between predicted confidence and the true binary outcome, capturing both absolute calibration at the individual level and the discriminative ability.

**Alternative prompting strategies and setups** To ensure the robustness of our findings, we experiment with a wide range of alternative prompting strategies and setups, and provide the results in Appendix A.2. (1) We explore different confidence expression styles such as providing only a linguistic descriptor without a probability, or outputting a numerical probability directly without binning. (2) Since some reasoning models perform two rounds of inference due to the forced CONFIDENCE REASONING, we also evaluate a similar two-step sequential prompting setup for non-reasoning

---

[3]We discuss the results on different of model sizes in Section 5.

[4]The Non-thinking mode is enabled by injecting an empty thinking block, `<think></think>`, at the beginning of the model's response.

[5]To elaborate, their thinking process mostly only contains SOLUTION REASONING, although they consistently do output their CONFIDENCE VERBALIZATION outside of the thinking termination token `</think>` without CoT.

Table 1: Benchmark result on knowledge-focused datasets. Reasoning models are highlighted in blue, and the best performance within each backbone LLM group is shown in **bold**. Lower ECE and Brier Scores, and higher AUROC, indicate better calibration. Accuracy is included for reference.

| Model | TriviaQA | | | | NonambigQA | | | |
|---|---|---|---|---|---|---|---|---|
| | Acc. | ECE↓ | Brier↓ | AUROC↑ | Acc. | ECE↓ | Brier↓ | AUROC↑ |
| **Qwen2.5-32B** | | | | | | | | |
| Qwen2.5-Inst. | 0.718 | 0.129 | 0.176 | 0.769 | 0.517 | 0.297 | 0.303 | 0.720 |
| R1-Distill-Qwen | 0.727 | **0.042** | 0.157 | 0.782 | 0.491 | **0.195** | **0.241** | 0.749 |
| OR1-Preview | 0.738 | 0.052 | 0.150 | 0.795 | 0.470 | 0.219 | 0.247 | **0.759** |
| QwQ | 0.768 | 0.063 | **0.137** | **0.807** | 0.535 | 0.226 | 0.250 | 0.757 |
| **GLM-4-32B-0414** | | | | | | | | |
| GLM-4-0414 | 0.814 | 0.084 | 0.137 | 0.675 | 0.640 | 0.246 | 0.269 | 0.643 |
| GLM-Z1-0414 | 0.824 | **0.029** | **0.120** | **0.777** | 0.570 | **0.209** | **0.251** | **0.721** |
| **EXAONE-3.5-32B** | | | | | | | | |
| EXAONE-3.5-Inst. | 0.715 | 0.130 | 0.178 | 0.749 | 0.511 | 0.302 | 0.302 | 0.721 |
| EXAONE-Deep | 0.687 | **0.104** | **0.175** | **0.763** | 0.452 | **0.288** | **0.289** | **0.743** |
| **Qwen3-32B** | | | | | | | | |
| Qwen3 Non-thinking | 0.711 | 0.207 | 0.230 | 0.650 | 0.511 | 0.403 | 0.403 | 0.572 |
| Qwen3 Thinking | 0.768 | **0.063** | **0.137** | **0.807** | 0.535 | **0.226** | **0.250** | **0.757** |

models. (3) We evaluate more advanced prompting strategies for non-reasoning models, as suggested by prior work, such as Top-K and Multi-step prompting [40, 45]. (4) While our main experiments use greedy decoding to avoid randomness and ensure reproducibility, we also test sampling-based decoding to confirm that our results generalize across decoding strategies. Across all setups, our findings remain consistent: reasoning models exhibit better calibration than their non-reasoning counterparts.

## 3.2 Experiment result

We observe that reasoning models consistently outperform their non-reasoning counterparts across all calibration metrics on knowledge-focused datasets (Table 1). Notably, reasoning models exhibit superior calibration even in cases where they underperform non-reasoning models in task accuracy. For example, GLM-Z1-0414 achieves 0.07 lower accuracy than GLM-4-0414 on NonambigQA, yet it obtains a 0.037 lower ECE and a 0.078 higher AUROC. This indicates that accurate confidence estimation is not merely a byproduct of better task performance. Additionally, this highlights a potential motivation for using reasoning models even on simple factual question-answering tasks: although they may not necessarily achieve higher accuracy, they communicate their confidence more reliably.

Interestingly, Qwen3 exhibits a substantial difference in calibration depending on whether it engages in CoT through Thinking or Non-thinking Mode. As Qwen3 is a single checkpoint, this suggests that the calibration observed in reasoning models primarily stems from their slow thinking CoT, rather than some other inherent differences between the models. We further support this claim in Section 4.3, where we show that even non-reasoning models can improve calibration by engaging in slow thinking through in-context learning.

Figure 2 (**left**) illustrates the superior calibration of R1-Distill-Qwen over Qwen2.5-Instruct on TriviaQA. Surprisingly, for confidence levels above 55%,[6] R1-Distill-Qwen exhibits near-perfect calibration, where its estimated confidence closely matches the actual accuracy (i.e., points lie near the line $y = x$, where $y$ is accuracy and $x$ is confidence). Plotting the sample frequency of the confidence bins (**right** of Figure 2) reveals that Qwen2.5-Instruct is highly overconfident, estimating either the 85% or 95% confidence bin for more than 80% of the total samples. In contrast, R1-Distill-Qwen produces estimations across a relatively diverse range of confidence levels, indicating less tendency to be overconfident. Nonetheless, there is room for improvement for R1-Distill-Qwen as well, as it rarely estimates confidence lower than 55%.

---

[6]55% corresponds to the "Better than even (0.5–0.6)" bin. Throughout the paper, we denote each bin by the average value of its numerical confidence range for brevity.

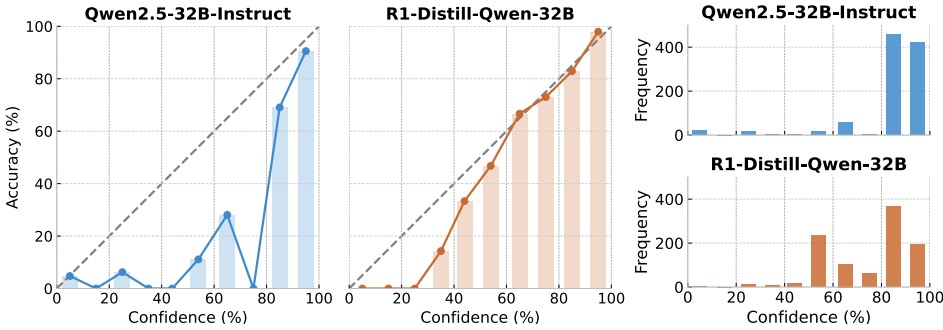

Figure 2: Accuracy (**left**) and sample frequency (**right**) across confidence bins for Qwen2.5-32B-Instruct and R1-Distill-Qwen-32B on TriviaQA.

Reasoning models are also better calibrated on reasoning-intensive datasets (Table 2), showing that our findings generalize to more complex tasks. However, we observe a few cases where non-reasoning models outperform their reasoning counterparts in AUROC; specifically, GLM-4-0414 on both Math subsets, and EXAONE-3.5-Instruct on the Non-Math subset of SuperGPQA. Upon closer inspection, we find that the root cause lies in the multiple-choice format of MMLU-Pro and SuperGPQA, which implicitly provides models with cues about their correctness based on the available options. Non-reasoning models, exhibiting overconfidence, tend to select the 95% confidence bin when their answers appear among the multiple-choice options, but when they do not, the models select the closest available option and assign a lower confidence of 85%. Relying on just two confidence bins, reinforced by implicit cues from the multiple-choice options, gives non-reasoning models an unearned advantage in discriminative power, which inflates their AUROC scores. Nonetheless, the non-reasoning models receive worse Brier Scores, as the metric, unlike AUROC, directly penalizes mismatches between predicted confidence and actual outcomes.

Table 2: Benchmark results on reasoning-intensive datasets. The full result with ECE and Accuracy is available in Appendix A.3.

| | SuperGPQA | | | | MMLU-Pro | | | |
| | Math | | Non-Math | | Math | | Non-Math | |
| Model | Brier↓ | AUROC↑ | Brier↓ | AUROC↑ | Brier↓ | AUROC↑ | Brier↓ | AUROC↑ |
|---|---|---|---|---|---|---|---|---|
| **Qwen-2.5-32B** | | | | | | | | |
| Qwen2.5-Inst. | 0.295 | 0.621 | 0.416 | 0.522 | 0.170 | 0.806 | 0.283 | 0.636 |
| R1-Distill-Qwen | 0.218 | **0.677** | 0.305 | 0.537 | 0.121 | 0.842 | 0.213 | 0.654 |
| OR1-Preview | **0.212** | 0.647 | **0.275** | **0.593** | 0.107 | 0.824 | **0.200** | 0.679 |
| QwQ | 0.217 | 0.664 | 0.314 | 0.581 | **0.094** | **0.871** | 0.222 | **0.687** |
| **GLM-4-32B-0414** | | | | | | | | |
| GLM-4-0414 | 0.256 | **0.681** | 0.459 | 0.516 | 0.131 | **0.794** | 0.282 | 0.626 |
| GLM-Z1-0414 | **0.216** | 0.633 | **0.348** | **0.572** | **0.095** | 0.783 | **0.222** | **0.648** |
| **EXAONE-3.5-32B** | | | | | | | | |
| EXAONE-3.5-Inst. | 0.345 | 0.590 | 0.441 | **0.567** | 0.243 | 0.676 | 0.324 | 0.590 |
| EXAONE-Deep | **0.261** | **0.645** | **0.385** | 0.542 | **0.110** | **0.776** | **0.261** | **0.648** |
| **Qwen3-32B** | | | | | | | | |
| Qwen3 Non-thinking | 0.296 | 0.588 | 0.440 | 0.552 | 0.155 | 0.708 | 0.285 | 0.632 |
| Qwen3 Thinking | **0.217** | **0.664** | **0.314** | **0.581** | **0.094** | **0.871** | **0.222** | **0.687** |

## 4 Analysis: slow thinking enables accurate confidence adjustments

In this section, we present an in-depth analysis showing that the enhanced calibration of reasoning models stems from slow thinking, which enables LLMs to dynamically adjust their confidence throughout the course of reasoning, as illustrated in Figure 1. First, we show that reasoning models *gradually* produce more accurate confidence estimates as their CoT unfolds, whereas non-reasoning models exhibit no such trend (Section 4.1). We then perform an ablation study, systematically removing components of the CoT, and identify that slow thinking behaviors, specifically exploring

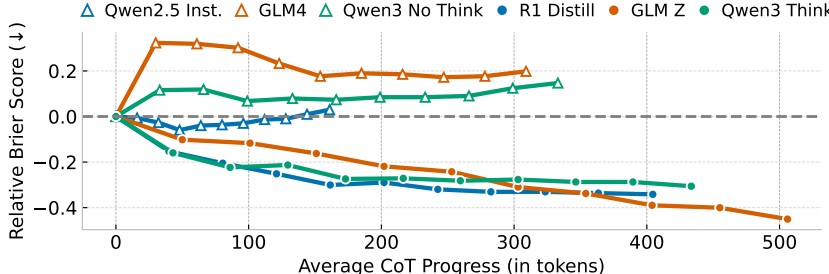

Figure 3: Relative change in Brier Score as CoT progresses on NonambigQA. Non-reasoning models are represented by triangles (△), and reasoning models by circles (●), with each model pair shown in matching colors.

alternatives and backtracking, are key contributors to improved calibration (Section 4.2). Finally, we demonstrate that simply prompting non-reasoning models to perform slow thinking through in-context learning yields similar benefits, supporting the notion that these gains are driven by slow thinking itself rather than other inherent differences (Section 4.3).

## 4.1 (Only) Reasoning models gradually get better calibrated as CoT progresses

If reasoning models are truly capable of dynamically adjusting and correcting their confidence throughout the CoT process, we would expect to observe a *steady, gradual* increase (rather than random or inconsistent patterns) in calibration over the course of reasoning. To test this, we collect the CoTs of both reasoning and non-reasoning models, and divide each instance into 11 cumulative segments, where each segment includes the first 0%, 10%, ..., up to 100% of the total number of generated tokens. We then prompt the models with each cumulative segment, appending tokens to trigger early termination of reasoning,[7] in order to collect the model's answer and its confidence expression at that point. We use the knowledge-focused datasets, TriviaQA and NonambigQA, as the models' accuracy remains relatively stable even when the reasoning process is terminated early.

Figure 3 shows the relative change in Brier Score compared to the initial value at 0 token length (i.e., before any CoT reasoning) on NonambigQA. We observe that all three reasoning models show progressively better calibration with the slope trending downwards. Surprisingly, for non-reasoning models, the Brier Score is the lowest without any CoT, and calibration worsens by the end of the CoT, with no consistent trend observed in between.

To quantify this trend across both datasets and all three metrics, we fit a linear model to each calibration metric over CoT progress and report the slope and its statistical significance across both datasets (Table 3). For reasoning models, we observe strong linear relationships in most cases (except for Qwen3 Thinking's ECE) with fitted slopes indicating steadily improving calibration over the course of the CoT. Meanwhile, non-reasoning models generally fail to yield statistically significant linear trends, and in the some cases where a statistically significant fit is observed, the slope surprisingly indicates worsening calibration.

## 4.2 Ablation study: exploring alternatives and refining matters the most

To determine which aspects are the most responsible for calibration gains, we analyze R1-Distill-Qwen-32B by removing individual components from its CoT. Specifically, we use the model's original CoTs on the knowledge-focused datasets, TriviaQA and NonambigQA, and re-prompt the model with targeted components removed. We ablate the following:

- **Confidence Reasoning**: We retain only the portion of the CoT that appears before the CONFIDENCE REASONING force prompt ("Okay, now let's assess my overall thinking process . . . ") described in Section 3.1, thereby ablating the effect of explicit reasoning about confidence.
- **Epistemic Markers**: Reasoning models frequently generate epistemic phrases that signal uncertainty, such as "I think" or "maybe", which could affect the confidence estimation. Therefore, we

---

[7] `</think>\nAnswer:` for reasoning models and `\nAnswer:` for non-reasoning models.

Table 3: Slopes of calibration metrics measured over CoT progress. **Bold Blue** indicates a steady improvement in calibration as CoT progresses, and **Bold Red** indicates a steady degradation. Bold colors are shown only when the trend is statistically significant ($p < 0.05$).

| Model | TriviaQA | | | | NonambigQA | | | |
|---|---|---|---|---|---|---|---|---|
| | Acc. | ECE↓ | Brier↓ | AUROC↑ | Acc. | ECE↓ | Brier↓ | AUROC↑ |
| **Qwen2.5-32B** | | | | | | | | |
| Qwen2.5-Inst. | 0.000 | **0.004** | 0.000 | **0.002** | 0.002 | **0.002** | 0.001 | **0.003** |
| R1-Distill-Qwen | 0.003 | **-0.007** | **-0.004** | **0.008** | 0.003 | **-0.015** | **-0.010** | **0.010** |
| **GLM-4-32B-0414** | | | | | | | | |
| GLM-4-0414 | 0.007 | -0.003 | **-0.003** | 0.000 | 0.005 | 0.002 | -0.001 | 0.003 |
| GLM-Z1-0414 | 0.009 | **-0.016** | **-0.011** | **0.026** | 0.009 | **-0.022** | **-0.020** | **0.022** |
| **EXAONE-3.5-32B** | | | | | | | | |
| EXAONE-3.5-Inst. | -0.001 | 0.001 | 0.000 | 0.002 | -0.001 | **0.004** | **0.003** | -0.001 |
| EXAONE-Deep | 0.006 | **-0.016** | **-0.011** | **0.026** | 0.002 | **-0.016** | **-0.016** | **0.024** |
| **Qwen3-32B** | | | | | | | | |
| Qwen3 Non-thinking | 0.003 | **0.004** | 0.001 | -0.001 | 0.004 | 0.002 | 0.002 | **-0.004** |
| Qwen3 Thinking | 0.004 | -0.005 | **-0.004** | **0.005** | 0.004 | -0.005 | **-0.004** | **0.005** |

remove or paraphrase such phrases, while preserving the rest of the content as closely as possible to the original.

- **Non-linear Reasoning**: Reasoning models follow complex non-linear reasoning paths, including behaviors such as exploring alternatives, refining, and backtracking. We prune these non-linear traces, retaining only the parts that directly support the model's final answer and thus making the reasoning path linear.

We also report the **No CoT** setting as a lower bound for reference, in which the model's reasoning is terminated immediately by prompting "`</think>`".

For both **Epistemic Markers** and **Non-linear Reasoning** removal, we use GPT-4.1 [29] guided by detailed instructions and three-shot demonstrations created by the authors. Only the portion of the CoT before the forced CONFIDENCE REASONING prompt is provided. For all three settings, the authors manually inspect 100 examples using predefined criteria, observing that more than 90% meet the intended requirements for each setting. Additional details on the experimental setup, including the full prompts used with GPT-4.1, the manual inspection process, and before-and-after examples of the CoTs, are provided in Appendix B.3.

Table 4: Ablation study on the CoT of R1-Distill-Qwen-32B. The "–" symbol indicates that the corresponding component is removed from the original CoT. The *No CoT* setting is included as a lower-bound reference.

| Method | TriviaQA | | | | NonambigQA | | | |
|---|---|---|---|---|---|---|---|---|
| | Acc. | ECE↓ | Brier↓ | AUROC↑ | Acc. | ECE↓ | Brier↓ | AUROC↑ |
| Original | 0.727 | 0.042 | 0.157 | 0.782 | 0.491 | 0.195 | 0.241 | 0.749 |
| – Confidence Reason. | 0.722 | 0.063 | 0.166 | 0.763 | 0.494 | 0.187 | 0.247 | 0.718 |
| – Epistemic Markers | 0.726 | 0.109 | 0.154 | 0.837 | 0.500 | 0.262 | 0.263 | 0.784 |
| – Non-linear Reason. | 0.731 | 0.161 | 0.179 | 0.734 | 0.500 | 0.341 | 0.320 | 0.728 |
| No CoT | 0.697 | 0.150 | 0.206 | 0.689 | 0.457 | 0.359 | 0.365 | 0.618 |

Table 4 presents the ablation results. First, we observe that **Confidence Reasoning** has only a minor effect on calibration, suggesting that the model's ability to express confidence accurately primarily stems from slow thinking about the question itself, rather than explicitly reasoning about its own confidence. Interestingly, removing **Epistemic Phrases** leads to a noticeable degradation in ECE, but results in improved AUROC. Upon further investigation, we find that the model becomes overconfident, yet surprisingly retains its ability to discriminate between correct and incorrect answers. Specifically, on TriviaQA, the model tends to predict either 95% or 65% confidence almost exclusively, as opposed to the more varied distribution in the original (see **right** of Figure 2). Despite this overconfidence, predictions at 95% are generally correct, while those at 65% are not, preserving the model's discriminative capacity. Restricting predictions to just two confidence bins, instead of distributing across a wider range, results in an advantage over the original model in AUROC.

Finally, our results show that **Non-linear Reasoning** has the greatest impact on calibration. When the reasoning path is constrained to a linear trajectory, all three metrics degrade, indicating a decline not only in absolute calibration but also in the model's ability to discriminate between correct and incorrect answers. Overall, our ablation results align with the observations in Section 4.1, indicating that as the CoT unfolds, the model gains more opportunity to express epistemic phrases, consider alternatives, and refine its reasoning which all contributes to improved calibration.

### 4.3   Non-reasoning models also better express their confidence with slow thinking

So far, we have shown that reasoning models are better calibrated (Section 3), that their calibration improves as the CoT unfolds (Section 4.1), and identified components in slow thinking that contribute to this improvement (Section 4.2). Yet an important question remains: *Are calibration gains from slow thinking exclusive to reasoning models that went through extensive fine-tuning or reinforcement learning with specialized reasoning data?*

In this section, we demonstrate that non-reasoning models also "better express their confidence" by simply engaging in slow thinking via in-context learning without any parameter updates. Specifically, we collect three random examples of R1-Distill-Qwen-32B slow thinking on held-out TriviaQA samples, and use them as few-shot exemplars to prompt non-reasoning models to reason in a similar manner.

Table 5 shows that non-reasoning models also consistently exhibit improved calibration across both datasets and all three metrics when prompted to engage in slow thinking. This highlights that the calibration gains in reasoning models, stem from the act of slow thinking itself rather than from inherent properties exclusive to reasoning models. This claim is further supported by our findings with Qwen3 in Section 3, where the model demonstrates significantly better calibration when performing CoT in Thinking mode compared to Non-Thinking mode.

Table 5: Benchmark results on non-reasoning models prompted to engage in slow thinking via in-context learning.

| | TriviaQA | | | | NonambigQA | | | |
|---|---|---|---|---|---|---|---|---|
| Model | Acc. | ECE↓ | Brier↓ | AUROC↑ | Acc. | ECE↓ | Brier↓ | AUROC↑ |
| Qwen2.5-32B-Inst. | 0.709 | 0.135 | 0.192 | 0.714 | 0.511 | 0.309 | 0.321 | 0.658 |
| + Slow Thinking | 0.709 | **0.091** | **0.167** | **0.784** | 0.481 | **0.269** | **0.290** | **0.696** |
| GLM-4-32B-0414 | 0.783 | 0.150 | 0.180 | 0.631 | 0.596 | 0.338 | 0.347 | 0.573 |
| + Slow Thinking | 0.800 | **0.029** | **0.126** | **0.796** | 0.622 | **0.126** | **0.218** | **0.715** |
| EXAONE-32B-Inst. | 0.728 | 0.118 | 0.183 | 0.712 | 0.519 | 0.326 | 0.331 | 0.670 |
| + Slow Thinking | 0.708 | **0.096** | **0.165** | **0.798** | 0.500 | **0.279** | **0.282** | **0.759** |

## 5   Discussion

**Does forcing longer CoT lead to improved calibration?**   Our findings in Section 4.1 suggest that reasoning models become better calibrated the longer they engage in CoT. This prompted us to test whether calibration can be further improved by forcing reasoning models to slow think longer. Specifically, we apply *budget forcing* [25] by appending "Wait," to the end of the CoT and conducting additional rounds of inference. Figure 4 (**left**) of Appendix A.1 shows that budget forcing does not necessarily lead to further improvements in calibration, suggesting that calibration gains likely stem from the quality rather than the quantity of slow thinking.

**Calibration scales better with model size in reasoning models**   Since our experiments are conducted on 32B-scale models, we further examine how the effects of slow thinking vary across different model scales. Figure 4 (**right**) of Appendix A.1 shows the relative calibration gain of reasoning models over their non-reasoning counterparts across various model scales. We find that the calibration gap widens as model size increases, suggesting that the benefits of slow thinking become more pronounced in larger, more capable LLMs. This trend is encouraging, as it suggests that slow thinking will become an increasingly effective mechanism for calibration as LLMs continue to improve.

**Room for improvement even for reasoning models** Generally, we observe that reasoning models still prefer to express high confidence, assigning values below 55% infrequently, as shown in Figure 2 (**right**). This tendency is further reflected in the calibration metrics: the lower task accuracy on NonambigQA leads to notably higher ECE and Brier Scores compared to TriviaQA (Table 1). While these observations suggest that the challenge of expressing uncertainty remains, our overall results demonstrate that slow thinking meaningfully improves calibration and represents a valuable direction for future model development.

## 6 Conclusion

In this work, we present a comprehensive study of reasoning models' ability to express confidence through their own words. Across diverse datasets and model families, we show that reasoning models are consistently better calibrated than their non-reasoning counterparts. Through detailed analysis, we trace these calibration gains to slow thinking behaviors which allow models to dynamically adjust their confidence throughout the chain-of-thought. Overall, our findings reveal that slow thinking offers more than just improved problem-solving: it also enhances the trustworthiness and reliability of LLMs by enabling them to better "know what they know."

## Acknowledgments

This work was supported by the InnoCORE program of the Ministry of Science and ICT (N10250156, 50%) and the Institute of Information & communications Technology Planning & Evaluation (IITP) grant funded by the Korea government (MSIT) (No.2022-0-00113, Developing a Sustainable Collaborative Multi-modal Lifelong Learning Framework, 25%; No.RS-2021-II212068, Artificial Intelligence Innovation Hub, 25%).

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

# A  Additional Experiment Results and Details

## A.1  Experiment Results in Discussion

In this section, we present the results referenced in the Discussion (Section 5). Specifically, Figure 4 shows the effect of forcing reasoning models to produce longer CoTs (**left**) and the effect of model scale on calibration (**right**).

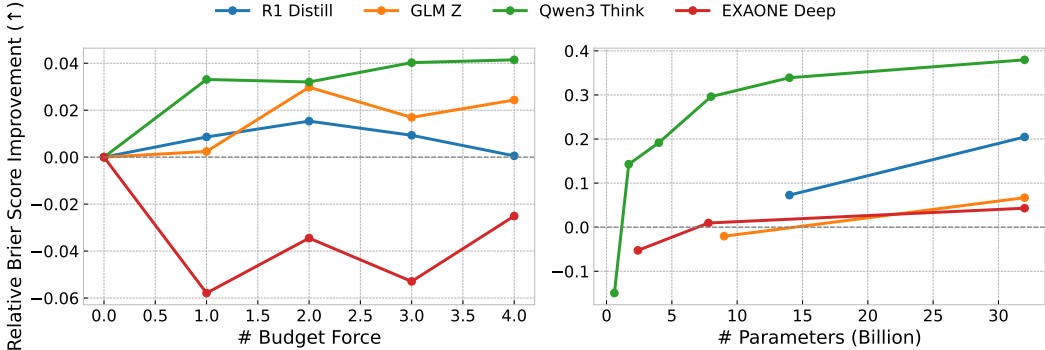

Figure 4: Relative change in Brier Score on NonambigQA under budget forcing (**left**) and across different model scales (**right**).

## A.2  Expanded results for robustness

In this section, we explore alternative setups and conditions to assess the robustness of our findings.

### A.2.1  Alternative Prompting Method for Non-reasoning Models

Table 6: Benchmark results with alternative prompting method for non-reasoning models.

| Model | TriviaQA | | | | NonambigQA | | | |
|---|---|---|---|---|---|---|---|---|
| | Acc. | ECE↓ | Brier↓ | AUROC↑ | Acc. | ECE↓ | Brier↓ | AUROC↑ |
| **Qwen2.5-32B** | | | | | | | | |
| Qwen2.5-Inst. | 0.718 | 0.129 | 0.176 | 0.769 | 0.517 | 0.297 | 0.303 | 0.720 |
| +Two-turn | 0.709 | 0.135 | 0.192 | 0.714 | 0.511 | 0.309 | 0.321 | 0.658 |
| +Top-K | 0.725 | 0.141 | 0.168 | 0.780 | 0.504 | 0.327 | 0.320 | 0.698 |
| +Multi-step | 0.734 | 0.107 | 0.177 | 0.735 | 0.524 | 0.278 | 0.305 | 0.659 |
| R1-Distill-Qwen | 0.727 | **0.042** | **0.157** | **0.782** | 0.491 | **0.195** | **0.241** | **0.749** |
| **GLM-4-32B-0414** | | | | | | | | |
| GLM-4-0414 | 0.814 | 0.084 | 0.137 | 0.675 | 0.640 | 0.246 | 0.269 | 0.643 |
| +Two-turn | 0.783 | 0.150 | 0.180 | 0.631 | 0.596 | 0.338 | 0.347 | 0.573 |
| +Top-K | 0.764 | 0.139 | 0.167 | 0.737 | 0.554 | 0.319 | 0.318 | 0.695 |
| +Multi-step | 0.798 | 0.105 | 0.153 | 0.652 | 0.667 | 0.228 | 0.262 | 0.589 |
| GLM-Z1-0414 | 0.824 | **0.029** | **0.120** | **0.777** | 0.570 | **0.209** | **0.251** | **0.721** |
| **EXAONE-3.5-32B** | | | | | | | | |
| EXAONE-3.5-Inst. | 0.715 | 0.130 | 0.178 | 0.749 | 0.511 | 0.302 | 0.302 | 0.721 |
| +Two-turn | 0.728 | 0.118 | 0.183 | 0.712 | 0.519 | 0.326 | 0.331 | 0.670 |
| +Top-K | 0.705 | 0.160 | 0.187 | 0.771 | 0.487 | 0.372 | 0.360 | 0.693 |
| +Multi-step | 0.685 | 0.115 | 0.186 | 0.728 | 0.501 | 0.269 | 0.286 | 0.698 |
| EXAONE-Deep | 0.687 | **0.104** | **0.175** | **0.763** | 0.452 | **0.288** | **0.289** | **0.743** |
| **Qwen3-32B** | | | | | | | | |
| Qwen3 Non-thinking | 0.711 | 0.207 | 0.230 | 0.650 | 0.511 | 0.403 | 0.403 | 0.572 |
| +Two-turn | 0.713 | 0.202 | 0.226 | 0.674 | 0.500 | 0.402 | 0.397 | 0.620 |
| +Top-K | 0.701 | 0.219 | 0.236 | 0.637 | 0.502 | 0.415 | 0.410 | 0.590 |
| +Multi-step | 0.711 | 0.175 | 0.208 | 0.737 | 0.501 | 0.369 | 0.367 | 0.634 |
| Qwen3 Thinking | 0.768 | **0.063** | **0.137** | **0.807** | 0.535 | **0.226** | **0.250** | **0.757** |

We apply alternative, more advanced prompting strategies to non-reasoning models to give them further advantage. Specifically:

Table 7: Benchmark results when providing only the linguistic descriptor.

| Model | TriviaQA | | | | NonambigQA | | | |
|---|---|---|---|---|---|---|---|---|
| | Acc. | ECE↓ | Brier↓ | AUROC↑ | Acc. | ECE↓ | Brier↓ | AUROC↑ |
| **Qwen2.5-32B** | | | | | | | | |
| Qwen2.5-Inst. | 0.714 | 0.099 | 0.170 | 0.756 | 0.516 | 0.259 | 0.279 | 0.705 |
| R1-Distill-Qwen | 0.707 | **0.059** | **0.169** | **0.762** | 0.480 | **0.219** | **0.255** | **0.736** |
| **GLM-4-32B-0414** | | | | | | | | |
| GLM-4-0414 | 0.820 | 0.077 | 0.131 | 0.713 | 0.639 | 0.222 | 0.252 | 0.672 |
| GLM-Z1-0414 | 0.833 | **0.080** | **0.118** | **0.777** | 0.584 | **0.143** | **0.226** | **0.719** |
| **EXAONE-3.5-32B** | | | | | | | | |
| EXAONE-3.5-Inst. | 0.722 | 0.116 | 0.165 | 0.760 | 0.514 | 0.295 | 0.293 | 0.729 |
| EXAONE-Deep | 0.689 | **0.108** | **0.161** | **0.805** | 0.457 | **0.302** | **0.287** | **0.768** |
| **Qwen3-32B** | | | | | | | | |
| Qwen3 Non-thinking | 0.717 | 0.169 | 0.213 | 0.673 | 0.510 | 0.370 | 0.376 | 0.600 |
| Qwen3 Thinking | 0.742 | **0.079** | **0.147** | **0.778** | 0.516 | **0.246** | **0.271** | **0.710** |

Table 8: Benchmark results when directly outputting a numerical probability without binning.

| Model | TriviaQA | | | | NonambigQA | | | |
|---|---|---|---|---|---|---|---|---|
| | Acc. | ECE↓ | Brier↓ | AUROC↑ | Acc. | ECE↓ | Brier↓ | AUROC↑ |
| **Qwen2.5-32B** | | | | | | | | |
| Qwen2.5-Instruct | 0.720 | 0.195 | 0.219 | 0.735 | 0.524 | **0.374** | 0.387 | 0.672 |
| R1-Distill-Qwen | 0.721 | **0.184** | **0.202** | **0.806** | 0.477 | 0.380 | **0.359** | **0.770** |
| **GLM-4-32B-0414** | | | | | | | | |
| GLM-4-0414 | 0.808 | 0.103 | 0.148 | 0.782 | 0.650 | **0.217** | **0.285** | 0.702 |
| GLM-Z1-0414 | 0.832 | **0.080** | **0.124** | **0.832** | 0.572 | 0.306 | 0.303 | **0.784** |
| **EXAONE-3.5-32B** | | | | | | | | |
| EXAONE-3.5-Instruct | 0.735 | 0.163 | 0.194 | 0.688 | 0.526 | 0.357 | 0.367 | 0.647 |
| EXAONE-Deep | 0.708 | **0.133** | **0.176** | **0.841** | 0.452 | **0.330** | **0.329** | **0.796** |
| **Qwen3-32B** | | | | | | | | |
| Qwen3 Non-Thinking | 0.707 | 0.238 | 0.264 | 0.739 | 0.509 | **0.379** | 0.448 | 0.659 |
| Qwen3 Thinking | 0.742 | **0.169** | **0.182** | **0.808** | 0.517 | 0.380 | **0.364** | **0.725** |

- **Two-turn**: Since some reasoning models perform two rounds of inference due to the forced CONFIDENCE REASONING (Section 3), we also evaluate a similar two-step sequential prompting setup for non-reasoning models. Specifically, we prompt the non-reasoning models to engage in SOLUTION REASONING in the first turn, followed by CONFIDENCE REASONING and CONFIDENCE VERBALIZATION in the second.

- **Top-K**: Following prior work, we prompt the model to generate $K$ candidate answers (with $K = 4$, as used in previous studies), each accompanied by a confidence estimate. The final answer is then selected as the one with the highest confidence.

- **Multi-step**: As suggested by prior work, we prompt the model to split its reasoning into $K$ steps (with $K = 4$), assessing its confidence after each step, and use the final confidence as the output.

Even when non-reasoning models are given additional support through alternative prompting methods, reasoning models still consistently outperform them (Table 6).

### A.2.2 Different Confidence Expression Styles

In this section, we investigate the effect of confidence expression styles. Specifically we test, (1) providing only a linguistic descriptor without a probability ("Almost Certain"), and (2) outputting a numerical probability directly without binning ("0.95").

Tables 7 and 8 present benchmark results when using only linguistic descriptors and when using numerical probabilities without bins, respectively. For using linguistic descriptors, reasoning models outperform non-reasoning models across all metrics and both datasets. In contrast, when models are prompted to output a numerical probability directly, we observe a general degradation in calibration for both reasoning and non-reasoning models. This finding is consistent with previous work suggesting that continuous probability expressions are less desirable for confidence estimation in LLMs [3]. Despite the inadequacy of this setting for reliable confidence estimation, reasoning models still generally outperform non-reasoning models.

### A.2.3 Sampling Instead of Greedy Decoding

Table 9: Benchmark results with sampling for R1-Distill-Qwen-32B. Result of Qwen2.5-32B-Instruct provided for reference. We conduct five runs and report the average and standard deviation of each metric.

| Model | TriviaQA | | | | NonambigQA | | | |
|---|---|---|---|---|---|---|---|---|
| | Acc. | ECE↓ | Brier↓ | AUROC↑ | Acc. | ECE↓ | Brier↓ | AUROC↑ |
| Qwen2.5-32B-Inst. | 0.718 | 0.129 | 0.176 | **0.769** | 0.517 | 0.297 | 0.303 | **0.720** |
| R1-Distill-Qwen-32B | | | | | | | | |
| - Average | 0.720 | **0.070** | **0.170** | 0.767 | 0.474 | **0.194** | **0.252** | 0.716 |
| - Standard Deviation | 0.002 | 0.001 | 0.001 | 0.001 | 0.001 | 0.001 | 0.000 | 0.001 |

While our main experiments use greedy decoding to reduce randomness and minimize computational cost, reasoning models are often recommended to be used with sampling. We therefore test whether they retain their superior calibration under these recommended decoding settings.

Table 9 shows that R1-Distill-Qwen-32B continues to outperform Qwen2.5-32B-Instruct in terms of ECE and Brier Score when using its recommended sampling configuration (temperature = 0.6). While its AUROC is slightly lower than that of Qwen2.5-32B-Instruct, further analysis reveals that this drop is due to R1-Distill-Qwen-32B producing a more diverse range of confidence values—unlike Qwen2.5-32B-Instruct, which mostly outputs just two bins. This diversity imposes a disadvantage in AUROC despite stronger absolute calibration performance.

### A.2.4 Bootstrapping

Table 10: Benchmark results under stochastic sampling (five bootstrap resamples). We report the average and standard deviation of each metric.

| Model | TriviaQA | | | | NonambigQA | | | |
|---|---|---|---|---|---|---|---|---|
| | Acc. | ECE↓ | Brier↓ | AUROC↑ | Acc. | ECE↓ | Brier↓ | AUROC↑ |
| Qwen2.5-32B-Inst. | | | | | | | | |
| - Average | 0.743 | 0.100 | 0.158 | 0.774 | 0.503 | 0.306 | 0.308 | 0.719 |
| - Standard Deviation | 0.019 | 0.017 | 0.012 | 0.022 | 0.010 | 0.007 | 0.005 | 0.010 |
| R1-Distill-Qwen-32B | | | | | | | | |
| - Average | 0.737 | **0.051** | **0.153** | **0.789** | 0.466 | **0.223** | **0.256** | **0.737** |
| - Standard Deviation | 0.010 | 0.014 | 0.006 | 0.019 | 0.017 | 0.021 | 0.010 | 0.009 |

To assess variability, we perform bootstrapping by generating five resampled subsets of 1,000 examples for each dataset and report the standard deviation across runs (Table 10). Due to the scale of our experiments, we do not run this analysis across all models and datasets, as doing so would incur unreasonable computational cost.

### A.3 Full Results on SuperGPQA and MMLU-Pro

In this section, we provide the full benchmark results on SuperGPQA (Table 11) and MMLU-Pro (Table 12), including both accuracy and ECE.

### A.4 Full CoT of the Qualitative Analysis

Figure 5 presents the full, untruncated CoT from R1-Distill-Qwen-32B shown in Figure 1.

Table 11: Full benchmark results on SuperGPQA.

| Model | Math | | | | Non-Math | | | |
|---|---|---|---|---|---|---|---|---|
| | Acc. | ECE↓ | Brier↓ | AUROC↑ | Acc. | ECE↓ | Brier↓ | AUROC↑ |
| **Qwen2.5-32B** | | | | | | | | |
| Qwen2.5-Inst. | 0.485 | 0.230 | 0.295 | 0.621 | 0.333 | 0.431 | 0.416 | 0.522 |
| R1-Distill-Qwen | 0.592 | 0.115 | 0.218 | **0.677** | 0.401 | **0.245** | 0.305 | **0.537** |
| OR1-Preview | 0.642 | **0.088** | **0.212** | 0.647 | 0.428 | **0.186** | **0.275** | 0.593 |
| QwQ | 0.658 | 0.118 | 0.217 | 0.664 | 0.448 | 0.256 | 0.314 | 0.581 |
| **GLM-4-32B-0414** | | | | | | | | |
| GLM-4-0414 | 0.609 | 0.208 | 0.256 | **0.681** | 0.402 | 0.460 | 0.459 | 0.516 |
| GLM-Z1-0414 | 0.669 | **0.112** | **0.216** | 0.633 | 0.402 | **0.327** | **0.348** | **0.572** |
| **EXAONE-3.5-32B** | | | | | | | | |
| EXAONE-3.5-Inst. | 0.343 | 0.336 | 0.345 | 0.590 | 0.238 | 0.494 | 0.441 | **0.567** |
| EXAONE-Deep | 0.571 | **0.205** | **0.261** | **0.645** | 0.309 | **0.396** | **0.385** | 0.542 |
| **Qwen3-32B** | | | | | | | | |
| Qwen3 Non-thinking | 0.604 | 0.257 | 0.296 | 0.588 | 0.412 | 0.442 | 0.440 | 0.552 |
| Qwen3 Thinking | 0.658 | **0.118** | **0.217** | **0.664** | 0.448 | **0.256** | **0.314** | **0.581** |

Table 12: Full benchmark results on MMLU-Pro.

| Model | Math | | | | Non-Math | | | |
|---|---|---|---|---|---|---|---|---|
| | Acc. | ECE↓ | Brier↓ | AUROC↑ | Acc. | ECE↓ | Brier↓ | AUROC↑ |
| **Qwen2.5-32B** | | | | | | | | |
| Qwen2.5-Inst. | 0.669 | 0.094 | 0.170 | 0.806 | 0.560 | 0.230 | 0.283 | 0.636 |
| R1-Distill-Qwen | 0.785 | 0.105 | 0.121 | **0.842** | 0.646 | 0.039 | 0.213 | 0.654 |
| OR1-Preview | 0.820 | **0.086** | 0.107 | 0.824 | 0.668 | **0.030** | **0.200** | 0.679 |
| QwQ | 0.833 | **0.047** | **0.094** | **0.871** | 0.648 | 0.137 | 0.222 | **0.687** |
| **GLM-4-32B-0414** | | | | | | | | |
| GLM-4-0414 | 0.802 | **0.067** | 0.131 | **0.794** | 0.637 | 0.250 | 0.282 | 0.626 |
| GLM-Z1-0414 | 0.861 | 0.072 | **0.095** | 0.783 | 0.669 | **0.124** | **0.222** | **0.648** |
| **EXAONE-3.5-32B** | | | | | | | | |
| EXAONE-3.5-Inst. | 0.598 | 0.156 | 0.243 | 0.676 | 0.498 | 0.285 | 0.324 | **0.590** |
| EXAONE-Deep | 0.809 | **0.036** | **0.110** | **0.776** | 0.580 | **0.191** | **0.261** | 0.648 |
| **Qwen3-32B** | | | | | | | | |
| Qwen3 Non-thinking | 0.787 | 0.083 | 0.155 | 0.708 | 0.626 | 0.249 | 0.285 | 0.632 |
| Qwen3 Thinking | 0.833 | **0.047** | **0.094** | **0.871** | 0.648 | **0.137** | **0.222** | **0.687** |

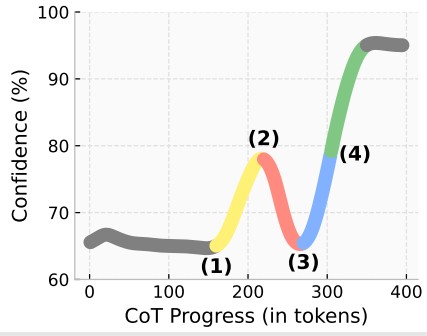

| | |
|---|---|
| **Question** | |

Who's the Athenian reformer that set up the council of 500?

| | |
|---|---|
| **Model CoT** | |

Okay, so I need to figure out which Athenian reformer set up the council of 500. Hmm, I remember that Athens had several important reformers, like Solon, Cleisthenes, and Pericles. Let me think about each of them.

Solon was an early reformer, right? He introduced some democratic elements, like the council of 400, I think. But wait, the question is about the council of 500. So maybe it wasn't Solon.

Then there's Cleisthenes. I recall that he was a significant figure in the development of Athenian democracy. He reorganized the political system after the overthrow of the tyrants. He introduced the concept of demes, which were local administrative units, and he increased the power of the Assembly. **(1)** Oh, and he also created the council of 500, which was called the Boule. Each deme would send a certain number of representatives to this council, which would meet regularly and handle day-to-day affairs. That makes sense because the council of 500 was a key part of the democratic reforms in Athens.

**(2)** Pericles came later, and he was more known for his leadership during the Golden Age of Athens, including the construction of the Parthenon and the expansion of democracy, **(3)** but I don't think he was the one who established the council of 500. That was already in place before his time.

**(4)** So putting it all together, Cleisthenes is the reformer who set up the council of 500. I'm pretty confident about this because it's a well-known part of his reforms. I don't think I'm mixing up any details here. The council of 500 was a major institution in the Athenian democracy, and Cleisthenes is credited with its creation as part of his efforts to decentralize power and make the government more representative.

Figure 5: Full CoT version of Figure 1.

# B Experimental Setup Details

## B.1 Experiments Compute Resources

We conduct our experiments on machines equipped with either Nvidia A6000 48GBs or A100 80GBs GPUs. For evaluating 32B-scale models, we use two GPUs. Leveraging vLLM [18] for efficient inference, evaluation of a 32B reasoning model on a single dataset takes approximately one hour on two A6000s.

## B.2 Additional Details on Inference Procedure

The full prompt given to the models to perform the three steps—SOLUTION REASONING, CONFIDENCE REASONING, and CONFIDENCE VERBALIZATION—along with the required answer and confidence formatting ("`Answer:ANSWER Confidence:CONFIDENCE`") is provided in Listing 1.

---
**Listing 1: Prompt used throughout our experiments**

First, reason through the question step by step to arrive at an answer.
Then, thoroughly assess your confidence in that answer by evaluating your thinking process so far.
Finally, classify your confidence into one of the following classes based on how likely your answer is to be correct:

- "Almost no chance" (0.0–0.1)
- "Highly unlikely" (0.1–0.2)
- "Chances are slight" (0.2–0.3)
- "Unlikely" (0.3–0.4)
- "Less than even" (0.4–0.5)
- "Better than even" (0.5–0.6)
- "Likely" (0.6–0.7)
- "Very good chance" (0.7–0.8)
- "Highly likely" (0.8–0.9)
- "Almost certain" (0.9–1.0)

Each category reflects the probability that your answer is correct.

At the very end of your output, format your answer and confidence as
**Answer**: $ANSWER
**Confidence**: $CLASS
where CLASS is one of the names (only the names without the probability ranges) of the classes above.

---

During our preliminary experiments, we occasionally observed cases—though rare—where models failed to produce the answer and confidence in the required format. To ensure robust extraction, we conduct an additional round of inference: for non-reasoning models, we request only the answer and confidence to be re-output in the specified format; for reasoning models, we first generate up to the `</think>` token, then replace it with `</think>Answer:` and perform a second round of inference.

For both reasoning and non-reasoning models, we use greedy decoding, with the maximum token length set to 4096 for knowledge-focused datasets and 8192 for reasoning-intensive datasets. The effect of using sampling is examined in Appendix A.2.

## B.3 Additional Details on Ablation Study

In this section, we detail the process for assessing the quality of the CoTs used in the ablation study in Section 4.2, along with before-and-after examples (Listing 2), and full prompts given to GPT-4.1 (Listing 3 and Listing 4). For quality check, we do the following:

- **Confidence Reasoning**: The authors manually inspect 100 CoT examples sampled from both datasets we find that only four examples contain explicit reasoning about the model's own confidence.

- **Epistemic Phrases**: The authors inspect 100 samples collected from both datasets and evaluate them based on two criteria: (1) whether epistemic phrases are absent, and (2) whether the remaining content stays true to the original. We find that 91 out of 100 examples satisfy both criteria.

- **Non-linear Reasoning**: We again provide detailed instructions along with three-shot examples. The authors then manually inspect 100 samples and evaluate them based on two criteria: (1) whether only the content supporting the final answer remains, and (2) whether the remaining content stays faithful to the original. We find that all 100 examples meet both criteria.

**Listing 2: Before-and-after examples of CoT**

**[Original (No Confidence Reasoning)]**
Okay, so I need to figure out which country the Order of the Elephant is the highest order of. I'm not too familiar with this, but I'll try to reason it out.

First, I remember that orders are usually honors or decorations given by countries to recognize people for their contributions. The Order of the Elephant sounds like a very prestigious award. I think it's associated with Denmark because I've heard of the Order of the Elephant in the context of Danish honors. But I'm not entirely sure.

Wait, maybe it's from another country. Let me think. I know the Order of the British Empire is from the UK, and the Order of Canada is from Canada. The Order of the Elephant doesn't ring a bell as being from those places. I think Denmark has a few orders, and the Order of the Elephant might be one of them. I'm pretty sure it's a Danish order, but I'm not 100% certain.

I should also consider if there are other countries with similar-sounding orders. Maybe Sweden or Norway? I don't recall any orders named Elephant from those countries. Denmark seems more likely because I've heard it mentioned in the context of their royal family or state honors.

So, putting it all together, I think the Order of the Elephant is the highest order of Denmark. I'm fairly confident, but I'm not 100% sure because I don't have direct knowledge. I might be mixing it up with another order, but I can't think of any others right now.

**[No Epistemic Phrases]**
Okay, so I need to figure out which country the Order of the Elephant is the highest order of. I'll try to reason it out.

First, orders are usually honors or decorations given by countries to recognize people for their contributions. The Order of the Elephant sounds like a prestigious award. It is associated with Denmark, and the Order of the Elephant appears in the context of Danish honors.

Wait, maybe it's from another country. The Order of the British Empire is from the UK, and the Order of Canada is from Canada. The Order of the Elephant does not appear to be from those places. Denmark has a few orders, and the Order of the Elephant is one of them. It is a Danish order.

Consideration can be given to other countries with similar-sounding orders, such as Sweden or Norway. However, there are no orders named Elephant from those countries. Denmark is more likely, as it is mentioned in the context of their royal family or state honors.

So, the Order of the Elephant is the highest order of Denmark.

**[No Non-linear Reasoning]**
Okay, so I need to figure out which country the Order of the Elephant is the highest order of.

Orders are honors given by countries, and the Order of the Elephant is associated with Denmark. It's mentioned in the context of the Danish royal family or state honors.

So, the Order of the Elephant is the highest order of Denmark.

**Listing 3: Prompt given to GPT-4.1 to remove Epistemic Phrases**

I will give you a model's thinking process to a question.
I want you to remove explicit expressions of uncertainty or confidence (like "I'm not sure", "I think", "maybe", "I'm a bit confused", "I'm pretty confident", etc.) in the model's thinking process.

Here are the details:
- Preserve the original thinking process and structure — do not delete any logical steps even if they seem mistaken or redundant.
- Rephrase uncertain sentences into neutral equivalents whenever possible. For example, change "I'm not sure if he wrote all three books or if someone else took over" into "He either wrote all three books or someone else took over."
- Preserve "Wait," whenever it appears. It signals a natural shift or reevaluation, not uncertainty.
- Don't shortcut the reasoning or fix factual errors.
- Keep the output as close as possible to the original wording, only editing where necessary to remove uncertainty.
- If the original thinking degenerates into repetition, or fails to reach a conclusion, stop the rewrite at the point where the reasoning becomes incomplete or broken. Do not invent, infer, or complete missing reasoning that was not present in the original.
- Strictly repeat the first sentence of the model's thinking process in the output.
- Use double linebreaks between paragraphs.

Here are some examples:

**Listing 4: Prompt given to GPT-4.1 to remove Non-linear Reasoning**

I will give you a model's thinking process to a question.
I want you to rewrite it into a concise and linear version, meaning the reasoning should move step-by-step directly toward the final answer without backtracking, or unnecessary side exploration. Also remove any expressions of uncertainty or confidence (like "I'm not sure", "I think", "maybe", "I'm a bit confused", "I'm pretty confident", etc.).

Here are the details:
- Linearize the reasoning — remove backtracking, "Wait," moments, or diversions.
- Keep only the details or examples directly supporting the final answer to make it concise.
- Remove any explicit expressions of uncertainty or confidence (such as "maybe", "I'm not sure", "I think", "I'm pretty confident", etc.).
- Do not fix any factual errors.
- Keep the tone casual and natural, like someone logically talking to themselves.
- If the original thinking degenerates into repetition, or fails to reach a conclusion, stop the rewrite at the point where the reasoning becomes incomplete or broken. Do not invent, infer, or complete missing reasoning that was not present in the original.
- Strictly repeat the first sentence of the model's thinking process in the output.
- Use double linebreaks between paragraphs.

Here are some examples:

# C  Licenses for existing assets

## C.1  Models

- R1-Distill [4]: MIT
- QwQ [35]: Apache-2.0
- OR1-Preview [11]: Apache-2.0
- Qwen-2.5 [33]: Apache-2.0
- Qwen-3 [34]: Apache-2.0
- GLM-4-0414 [39]: MIT
- GLM-Z1-0414 [39]: MIT
- EXAONE-3.5 [19]: EXAONE
- EXAONE-Deep [19]: EXAONE

## C.2  Datasets

- SuperGPQA [21]: ODC-BY, `https://huggingface.co/datasets/m-a-p/SuperGPQA`
- MMLU-Pro [42]: MIT, `https://huggingface.co/datasets/TIGER-Lab/MMLU-Pro`
- NonambigQA [17, 23]: CC-BY-SA-3.0, `https://github.com/shmsw25/AmbigQA`
- TriviaQA [15]: Apache-2.0, `https://huggingface.co/datasets/mandarjoshi/trivia_qa`

