# OpenReview forum: "Reasoning Models Better Express Their Confidence"
_NeurIPS.cc/2025/Conference — NeurIPS 2025 poster_

### Official Review · Reviewer_PPaK · 2025-06-14

**Clarity:** 3
**Significance:** 3
**Originality:** 3
**Rating:** 4
**Confidence:** 4

**Summary:**

This paper investigates the confidence calibration of reasoning and non-reasoning language models and finds that reasoning models consistently demonstrate better confidence calibration across a variety of datasets.
Further analysis shows that improvement stems from exploring alternatives and backtracking.
Based on this finding, this work propose a simple but effect few-shot method to calibrate the non-reasoning model.

**Questions:**

1. TriviaQA is a free-form QA dataset. How is answer correctness determined in this paper? Is it based on exact match, or model-based scoring? The method used to assess correctness should be clearly specified

**Ethical Concerns:**

["NO or VERY MINOR ethics concerns only"]

**Final Justification:**

The rebuttal addressed my concerns on the prompt.

**Limitations:**

yes

**Quality:**

2

**Strengths And Weaknesses:**

### Strengthness
1. The observation that reasoning models express confidence more effectively is  novel. The analysis provided is intuitive and has the potential to influence future developments in both confidence calibration and reasoning-model.
2. The analysis in Section 4.2 carefully disentangles multiple factors within the reasoning process that may affect calibration, ultimately identifying _non-linear reasoning_ as the dominant contributor. This insight is valuable and directly informs the design of the simple but highly effective calibration method proposed in Section 4.3.


### Weakness

1. Prompt Is Very Important! The same prompt is used for both reasoning and non-reasoning models, it appears to be specifically designed for reasoning models. [1] shows prompting non-reasoning models to output CoT or express confidence linguistically can actually degrade calibration. It is possible that the observed calibration advantage of reasoning models is prompt-specific. The authors should consider testing multiple prompt variants for non-reasoning models, including the most effective ones reported in [1].
2. The calibration gap are relatively small in some cases. For instance, in Table 10 (TriviaQA), the Brier score difference is only 0.005 and AUROC differs by 0.015. These are modest gains and may fall within the range of variation caused by prompt sensitivity alone.

\[1\] Just Ask for Calibration: Strategies for Eliciting Calibrated Confidence Scores from Language Models Fine-Tuned with Human Feedback

---

> ### Author Rebuttal · Authors · 2025-07-29
>
> Thank you Reviewer PPaK for your positive review of our work.
>
> We appreciate your recognition of its novelty, potential impact, and the value of the analysis.
>
> We address your "Weaknesses" in (W1) and (W2), and answer your question in (Q1).
>
> ---
>
> ### **(W1)**
> > Prompt Is Very Important! The same prompt is used for both reasoning and non-reasoning models, it appears to be specifically designed for reasoning models. [1] shows prompting non-reasoning models to output CoT or express confidence linguistically can actually degrade calibration. It is possible that the observed calibration advantage of reasoning models is prompt-specific. The authors should consider testing multiple prompt variants for non-reasoning models, including the most effective ones reported in [1].
>
> We respectfully believe that your concern rises from a misunderstanding.
>
> We would like to first clarify that we did not specifically design the prompt for reasoning models. Our prompt used in the main experiments was designed after the prompts used in previous seminal works (back when there were no reasoning models) [1, 2], and is generalizable for both reasoning and non-reasoning models. We will revise our manuscript to clearly convey this point.
>
> We would also like to clarify that we have already investigated the robustness of our findings across multiple different prompts, which includes the points that you have mentioned. This is stated in L:157 and in Appendix A.2.
>
> Specifically,
> - In Appendix A.2.1 we test advanced prompting strategies (Top-K and Multi-step) suggested by the work you have mentioned [1].
> - In Appendix A.2.2, we tested the effect of different confidence expression styles (linguistic, numeric, and linguistic + numeric).
>
> Indeed, calibration does show some sensitivity to prompt variations, as you noted. Yet, across all tested settings, our main finding remains consistent: reasoning models are better calibrated than non-reasoning models. Given the breadth of prompting strategies we evaluated, we believe our conclusion is well-supported and robust.
>
> If there are any additional aspects of prompt robustness you'd like us to explore, we're happy to provide further results or clarifications.
>
> ---
> ### **(W2)**
> > The calibration gap are relatively small in some cases. For instance, in Table 10 (TriviaQA), the Brier score difference is only 0.005 and AUROC differs by 0.015. These are modest gains and may fall within the range of variation caused by prompt sensitivity alone.
>
> We hope that we have sufficiently addressed the prompt sensitivity concern in (W1). Here we explain the reason behind the small calibration gap in TriviaQA in a few cases. In a nutshell, this is because LLMs achieve relatively high accuracy on TriviaQA (around ~70% accuracy), which makes the overconfidence and poor calibration of non-reasoning models less pronounced. In fact, on a more challenging dataset like NonambigQA (with ~50% accuracy) the calibration gap between reasoning and non-reasoning models widens, leading to a Brier score gap of 0.083 (Table 10).
>
> To elaborate, non-reasoning models tend to be overconfident, frequently assigning high confidence levels (e.g., 85% or 95%) regardless of correctness (see Figure 2). When task accuracy is also high, this overconfidence appears less problematic, as the confidence roughly aligns with accuracy. However, in lower-accuracy settings, this “unintended alignment” breaks, as the overconfidence is no longer aligned with the accuracy.
>
> On the other hand, reasoning models are able to generate relatively accurate confidence levels. This strength is more pronounced on low-accuracy tasks, where non-reasoning models do not have the unearned benefit of the “unintended alignment”.
>
> All in all, we argue that the small calibration gap in TriviaQA doesn’t necessarily suggest that reasoning and non-reasoning models show similar levels of calibration. Rather, the small gap reflects limitations of the calibration metrics in capturing overconfidence and poor calibration when accuracy is high.
>
> To further support our point, we provide additional results on SimpleQA [3], a factoid-based QA dataset similar to TriviaQA and NonambigQA but substantially more challenging.
>
> | Model           | Accuracy |   ECE   |  Brier  | AUROC |
> |----------------|----------|---------|---------|-------|
> | Qwen 3 No Think | 0.064    | 0.787   | 0.709   | 0.536 |
> | Qwen 3 Think    | 0.067    | **0.499** | **0.335** | **0.672** |
> |                |          |         |         |       |
> | GLM 4           | 0.097    | 0.676   | 0.599   | 0.619 |
> | GLM Z1          | 0.11     | **0.502** | **0.367** | **0.687** |
>
> As shown in the Table above, the models achieve low accuracy, which leads to a massive difference in calibration score between reasoning and non-reasoning models (0.374 and 0.232 Brier score difference for Qwen 3 and GLM, respectively), clearly highlighting the superior calibration of reasoning models.
>
> Thank you for raising this point. We will make sure to include this explanation in the Discussion section of the paper.
>
> ---
> ### **(Q1)**
> > TriviaQA is a free-form QA dataset. How is answer correctness determined in this paper? Is it based on exact match, or model-based scoring? The method used to assess correctness should be clearly specified
>
> Following prior work on verbalized confidence [1, 2], we also use an LLM to match the generated answer to the ground truth, as noted in L:143.
>
> Thank you again for reviewing our work. If we have overlooked any part of your concern, please let us know. We are happy to engage further.
>
> ---
> [1] Tian, Katherine, et al. "Just Ask for Calibration: Strategies for Eliciting Calibrated Confidence Scores from Language Models Fine-Tuned with Human Feedback." EMNLP 2023
>
> [2] Xiong, Miao, et al. "Can llms express their uncertainty? an empirical evaluation of confidence elicitation in llms." ICLR 2024
>
> [3] Wei, Jason, et al. "Measuring short-form factuality in large language models." arXiv preprint arXiv:2411.04368

---

> > ### Comment · Reviewer_PPaK · 2025-08-05
> >
> > Thank you for the clarification and the added experiments. I will keep the score since it is already positive. Good luck with the final decision!

---

> > > ### Author Response · Authors · 2025-08-05
> > >
> > > That's fair. Thank you for your response!

---

### Official Review · Reviewer_ofVM · 2025-06-29

**Clarity:** 3
**Significance:** 3
**Originality:** 3
**Rating:** 5
**Confidence:** 4

**Summary:**

This work presents a study showing how reasoning models are better calibrated. The study is conducted on multiple LLMs and four different datasets. When explicitly asked to reason about confidence in their thoughts, the reasoning models show better calibration ability according to three metrics (ECE, Brier, and AUROC) despite having similar accuracy. This study also contains ablations showing what aspects of the thoughts are helpful, and shows non-reasoning models can also demonstrate similar improvements by using the thoughts as in-context examples.

**Questions:**

1. How do you design the prompts that are used in this study? Have you tried multiple different prompts? Is the model sensitive to these changes?

**Ethical Concerns:**

["NO or VERY MINOR ethics concerns only"]

**Final Justification:**

This is overall a good empirical analysis paper. I don't see major concerns for this study. The main concern I had was on the sensitivity of the results w.r.t decoding methods and prompt designs, which the authors claim to be robust. I don't see other major concerns from the review/rebuttal from other reviewers, so I keep my original score.

**Limitations:**

yes

**Quality:**

3

**Strengths And Weaknesses:**

## Strengths
1. Understanding how reasoning models (or thinking models) change their behaviors compared to other models is an important topic.
2. I like the ablations studying what specific parts of the thought help calibration. The analysis is well-designed and insightful.
3. The experiment is comprehensive, covering multiple models and multiple datasets.

## Weaknesses
1. I'm fairly convinced that the results in this study are generalizable, but it would still be great to show multiple ways of confidence expression in this work. Currently, this work only shows results on verbalized confidence. It would be interesting to see if the same trends also hold if you sample the models multiple times and compute the confidence from samples. Additionally, I think with greedy decoding, it's probably very hard to sample low-confidence answers, and that's why in Figure 2, the lower part of that figure has very few samples.
2. This is a minor point. I'm slightly concerned about using "Wait" as the token to enforce longer CoT in Line 299. I don't feel "Wait" is semantically neutral and it can lead to unintended changes, hence hurting calibration. It would be better if the same study were implemented by just avoiding the tokens that end the thinking process.

---

> ### Author Rebuttal · Authors · 2025-07-29
>
> Thank you Reviewer ofVM for your positive review of our work.
>
> We especially appreciate your recognition of the importance of our topic and your acknowledgement of the thoroughness of our experiments and ablations.
>
> We address your "Weaknesses" in (W1) and (W2), and answer your question in (Q1).
>
> ---
>
> ### **(W1)**
> > I'm fairly convinced that the results in this study are generalizable, but it would still be great to show multiple ways of confidence expression in this work. Currently, this work only shows results on verbalized confidence. It would be interesting to see if the same trends also hold if you sample the models multiple times and compute the confidence from samples. Additionally, I think with greedy decoding, it's probably very hard to sample low-confidence answers, and that's why in Figure 2, the lower part of that figure has very few samples.
>
> While there are multiple different approaches to assess an LLM’s confidence (including the consistency based approach you have mentioned), each approach has its own specific challenges.
>
> The challenge of verbalized confidence stems from the inability of LLMs to accurately judge their own correctness (mostly due to overconfidence). In our work, we have shown the potential of slow thinking in overcoming this specific challenge. Consistency based confidence, on the other hand, does not face the same challenge as the LLM does not judge its own correctness. Rather the challenge rises from increased computation due to repeated sampling.
>
> We agree that exploring other confidence estimation approaches for reasoning models would be valuable future exploration. However, in this work, similar to prior studies that focus exclusively on verbalized confidence [1, 2], we chose to narrow our scope in order to tackle its unique challenges and provide a focused, in-depth analysis.
>
> Regarding the effect of greedy decoding, we also test sampling and report the average calibration scores with standard deviations in Appendix A.2.3, where our findings are consistent: reasoning models show better calibration even when using sampling.
>
> ---
>
> ### **(W2)**
> > This is a minor point. I'm slightly concerned about using "Wait" as the token to enforce longer CoT in Line 299. I don't feel "Wait" is semantically neutral and it can lead to unintended changes, hence hurting calibration. It would be better if the same study were implemented by just avoiding the tokens that end the thinking process.
>
> Thank you for raising an interesting point.
>
> We have chosen to use “Wait” to enforce longer CoTs following a previous study [3]. Specifically, the previous work has demonstrated “Wait”’s effectiveness over other alternatives. Indeed, it’s possible that the semantics of “Wait” may have some influence on calibration. However, we believe investigating the severity of this effect and further designing an alternative that both effectively forces longer reasoning and eliminates such influence is complex on its own and warrants a separate study.
>
> ---
>
> ### **(Q1)**
> > How do you design the prompts that are used in this study? Have you tried multiple different prompts? Is the model sensitive to these changes?
>
> Our prompt design follows prior work on verbalized confidence. Specifically, for our main experiments we have referenced the CoT prompting settings of seminal works [1, 2].
>
> As stated in L:157, we tested several alternative prompting strategies and reported the results in Appendix A.2. While we observed some calibration sensitivities, our main finding that reasoning models are better calibrated than non-reasoning models remains consistent across prompts.
>
> Thank you again for reviewing our work. Please let us know if anything is still unclear, we’d be happy to follow up.
>
> ---
>
> [1] Tian, Katherine, et al. "Just Ask for Calibration: Strategies for Eliciting Calibrated Confidence Scores from Language Models Fine-Tuned with Human Feedback." EMNLP 2023
>
> [2] Xiong, Miao, et al. "Can llms express their uncertainty? an empirical evaluation of confidence elicitation in llms." ICLR 2024
>
> [3] Muennighoff, Niklas, et al. "s1: Simple test-time scaling." arXiv preprint arXiv:2501.19393 (2025).

---

> > ### Comment · Reviewer_ofVM · 2025-08-06
> >
> > Thank you for your response. I will maintain my original rating of 5.

---

### Official Review · Reviewer_VJFc · 2025-07-01

**Clarity:** 3
**Significance:** 1
**Originality:** 1
**Rating:** 3
**Confidence:** 5

**Summary:**

This paper studies reasoning models' behavior on confidence and find that they not only perform better but also give more reliable confidence. They test six models on six tasks and see that reasoning models are better at expressing their confidence in almost all cases. The authors further identified that this is because reasoning models take time to think and try different ideas, which helps them adjust their confidence. This paper finally pointed out that the confidence gets worse for reasoning models without slow thinking, while non-reasoning models can improve confidence if they are guided to think step-by-step.

**Questions:**

Please see Weakness 2-4.

**Ethical Concerns:**

["NO or VERY MINOR ethics concerns only"]

**Final Justification:**

After reading the rebuttal and considering the discussions, I am updating my assessment based on the following:

W1: The authors differentiate their focus on verbalized confidence, not logit-based, and show that "slow thinking" CoT brings new insights into calibration.

W2: The authors claimed that calibration does not imply correctness.

W3: Additional experiments on SimpleQA support their claims that reasoning models do not underperform and show better calibration.

W4: They demonstrate that comparisons within the same checkpoint control for training differences.

This paper offers additional contributions to understanding confidence calibration in reasoning models.

**Limitations:**

Yes.

**Quality:**

2

**Strengths And Weaknesses:**

Strengths:

1. Exploring confidence of reasoning models is an important problem.

2. Comprehensive experiments and ablations to obtain the conclusion.

3. Writing is good and clear.

Weakness:

1. The finding that increased reasoning (CoT) correlates with higher confidence is not novel to this paper. Extensive prior research has established that CoT typically leads to enhanced confidence [1-3].

[1]. Wang, Xuezhi, and Denny Zhou. "Chain-of-thought reasoning without prompting." arXiv preprint arXiv:2402.10200 (2024).
[2]. Wang, Zezhong, et al. "Chain-of-Probe: Examing the Necessity and Accuracy of CoT Step-by-Step." arXiv preprint arXiv:2406.16144 (2024).
[3]. Foodeei, Darius, Simin Fan, and Martin Jaggi. "Semantic uncertainty in advanced decoding methods for LLM generation." arXiv preprint arXiv:2506.17296 (2025).

2. Examining solely the relationship between reasoning and confidence may be superficial. The connection between improved confidence and enhanced correctness remains unclear. Previous studies [4-5] have suggested that while long CoT reasoning can boost confidence, it may simultaneously reinforce biases and hallucinations, potentially leading to incorrect conclusions.

[4]. Lu, Haolang, et al. "Auditing Meta-Cognitive Hallucinations in Reasoning Large Language Models." arXiv preprint arXiv:2505.13143 (2025).
[5]. Cheng, Jiahao, et al. "Chain-of-Thought Prompting Obscures Hallucination Cues in Large Language Models: An Empirical Evaluation." arXiv preprint arXiv:2506.17088 (2025).

3. Reasoning models may perform significantly worse than non-reasoning models on some knowledge-dense benchmarks such as SimpleQA. It would be interesting to investigate that how reasoning could impact confidence and correctness on these benchmarks.

4. It would be beneficial to conduct more controlled training experiments to establish robust conclusions. For instance, since QwQ and R1-Distill-32B underwent post-training on large-scale and diverse datasets, directly comparing them with Qwen2.5-32B-Instruct may not constitute a fair experimental setup.

---

> ### Author Rebuttal · Authors · 2025-07-29
>
> Thank you Reviewer VJFc for your time and effort in reviewing our work.
>
> We greatly appreciate you acknowledging the importance of our topic and the value of our comprehensive experiments and ablations.
>
> We address your “Weakness” in (W1-W4).
>
> ---
>
> ### **(W1)**
>
> > The finding that increased reasoning (CoT) correlates with higher confidence is not novel to this paper. Extensive prior research has established that CoT typically leads to enhanced confidence.
>
> We believe that this concern mostly arises from a misunderstanding.
>
> We first want to kindly clarify that traditional CoT (as opposed to the slow thinking CoT of reasoning models) does not enhance calibration for verbalized confidence, which is the setting of our paper.
>
> While the works you have shared does indeed show that traditional CoT can lead to higher confidence, their focus has been on logit-based confidence estimation. In contrast, our work centers on verbalized confidence estimation which is conceptually distinct from logit-based confidence, in that it prompts LLMs to assess and express their confidence directly in their output and does not leverage internal states such as logit-based probabilities. Not only does the methodologies fundamentally differ, verbalized confidence estimation suffers from its own unique challenge: the overconfidence of LLMs [2].
>
> Crucially, according to prior research [1, 2], traditional CoT does not mitigate such overconfidence or enhance calibration for verbalized confidence. Our findings in Section 4.1 are consistent with this: we observe that non-reasoning models do not gain improved calibration with CoT. In contrast, we find that reasoning models get increasingly better calibrated with their slow thinking CoT. Therefore, our finding challenges the prior consensus that CoT does not enhance verbalized confidence calibration, contrary to your concern.
>
> Furthermore, we would like to kindly emphasize that the slow thinking CoT studied in our work is not just simply “increased CoT” already studied by prior works. The content of slow thinking differs considerably from traditional CoT, with complex behaviors such as backtracking or exploring alternatives [3]. The effect of such behaviors on calibration is not well explored by previous research.
>
> Finally, while reasoning models are becoming increasingly popular, their calibration remains underexplored compared to non-reasoning models. Our work is, to the best of our knowledge, the first to suggest that reasoning models exhibit superior calibration, providing a foundation for future research in this area.
>
> To summarize, we address your concern on the novelty as:
> - We demonstrate that slow thinking CoT enhances verbalized confidence calibration, challenging the prior understanding that CoT does not enhance verbalized confidence calibration (Section 4.1).
> - We provide one of the first comprehensive experimental analyses on the impact of slow thinking behaviors on calibration, highlighting their positive effect (Section 4.2).
> - To our knowledge, our work is the first to suggest that reasoning models exhibit enhanced calibration compared to non-reasoning models in verbalized confidence estimation (Section 3).
>
> We appreciate you sharing the related works. We will cite them as Related Work and clearly convey how our findings offer novel contributions beyond them.
>
> ---
>
> ### **(W2)**
>
> > Examining solely the relationship between reasoning and confidence may be superficial. The connection between improved confidence and enhanced correctness remains unclear. Previous studies [4-5] have suggested that while long CoT reasoning can boost confidence, it may simultaneously reinforce biases and hallucinations, potentially leading to incorrect conclusions.
>
> First, we respectfully ask you to consider that these findings only emerged recently and were not available at the time of our submission. Specifically, the two preprints you kindly shared were uploaded to arXiv after NeurIPS submission deadline (May 15th) and are not "previous studies". As such, we could not have reasonably incorporated or addressed them in our work. Given this, we hope our work will be considered within the scope of the knowledge available at the time of submission.
>
> That said, we are glad to discuss the point you’ve brought up in more detail.
>
> We would like to clarify that increase in calibration does not directly correlate to increase in correctness. Calibration (or confidence), in nature, is not intended as a means to enhance correctness; rather, it measures how well a model knows how likely they will be correct [4]. In other words, a well-calibrated model is better at estimating the correctness of their answer, but it is not necessarily better at actually arriving at the correct answer.
>
> Therefore, the findings of the preprints you cited regarding biases and hallucinations in reasoning models do not challenge our conclusion. A model may be better calibrated even if it (allegedly) is less accurate. To elaborate, even if a model frequently gives incorrect answers, it can still achieve high calibration if it precisely acknowledges that it might be wrong.
>
> Nonetheless, investigating why reasoning models allegedly produce more hallucinations despite their increased confidence is an interesting direction for future work. The fact that our work is already helping to raise such follow-up research questions when coupled with more recent works highlights its potential impact on shaping future research directions.
>
> Please let us know if we have misunderstood your point, we would be happy to further clarify or address it.
>
>
>
> ### **(W3)**
>
> > Reasoning models may perform significantly worse than non-reasoning models on some knowledge-dense benchmarks such as SimpleQA. It would be interesting to investigate that how reasoning could impact confidence and correctness on these benchmarks.
>
> Unfortunately, we were unable to find any references that support your comment that reasoning models may perform significantly worse on benchmarks such as SimpleQA. The official SimpleQA preprint reports comparable performance between the reasoning and non-reasoning models [5].
>
> In our paper, we have already evaluated reasoning and non-reasoning models on knowledge-based benchmarks, specifically TriviaQA and NonambigQA (Section 3.2). These are simple factoid QA tasks, similar to SimpleQA. On these datasets, we also did not observe that reasoning models perform significantly worse than non-reasoning models; in fact, both model types achieve comparable accuracy, while reasoning models demonstrate superior calibration.
>
> Nonetheless, we appreciate your inquiry and have conducted additional experiments on SimpleQA.
>
> | Model           | Accuracy |   ECE   |  Brier  | AUROC |
> |----------------|----------|---------|---------|-------|
> | Qwen 3 No Think | 0.064    | 0.787   | 0.709   | 0.536 |
> | Qwen 3 Think    | 0.067    | **0.499** | **0.335** | **0.672** |
> |                |          |         |         |       |
> | GLM 4           | 0.097    | 0.676   | 0.599   | 0.619 |
> | GLM Z1          | 0.11     | **0.502** | **0.367** | **0.687** |
>
> Consistent with our results on TriviaQA and NonambigQA, we find that reasoning models do not necessarily exhibit lower accuracy than non-reasoning models, yet they demonstrate significantly better calibration.
>
> ---
>
> ### **(W4)**
>
> > It would be beneficial to conduct more controlled training experiments to establish robust conclusions. For instance, since QwQ and R1-Distill-32B underwent post-training on large-scale and diverse datasets, directly comparing them with Qwen2.5-32B-Instruct may not constitute a fair experimental setup.
>
> We agree that directly comparing reasoning models like R1-Distill to models such as Qwen2.5-Instruct may involve confounding factors, as we do not fully know how these models differ in post-training.
>
> However, we would like to emphasize that we already provide experiments with this confounding factor fully controlled, since these experiments draw comparisons within a single checkpoint.
>
> - In our main experiments, we compare Qwen3 Thinking mode with Non-thinking mode, where we consistently observe that the Thinking mode is better calibrated (Section 3.2, L:177). Qwen3 is a hybrid model that supports both reasoning (Thinking) and non-reasoning (Non-thinking) modes.
>
> - In our analysis (Section 4.3), we further demonstrate that non-reasoning models can improve their calibration by engaging in slow thinking through in-context learning. Since the model checkpoint remains identical and the only difference is the presence of slow thinking, we believe this setup fully addresses the concern you raised.
>
> Therefore, while we acknowledge that a confounding factor may be present in certain settings, in the experiments where this factor was fully controlled, we observed consistent results. This demonstrates the robustness of our main findings and indicates that the confounding factor is not substantial enough to undermine them.
>
> We will revise the write-up to clearly convey this point.
>
> Thank you again for reviewing our work. Please let us know if any part of your concern remains unaddressed. We are more than happy to discuss further.
>
> ---
>
> [1] Tian, Katherine, et al. "Just Ask for Calibration: Strategies for Eliciting Calibrated Confidence Scores from Language Models Fine-Tuned with Human Feedback." EMNLP 2023
>
> [2] Xiong, Miao, et al. "Can llms express their uncertainty? an empirical evaluation of confidence elicitation in llms." ICLR 2024
>
> [3] Gandhi, Kanishk, et al. "Cognitive behaviors that enable self-improving reasoners, or, four habits of highly effective stars." arXiv preprint arXiv:2503.0130
>
> [4] Geng, Jiahui, et al. "A Survey of Confidence Estimation and Calibration in Large Language Models." NAACL 2024
>
> [5] Wei, Jason, et al. "Measuring short-form factuality in large language models." arXiv preprint arXiv:2411.04368

---

> > ### Comment · Reviewer_VJFc · 2025-08-05
> >
> > Thank you for the clarification and the added experiments. I will increase my score accordingly.

---

> > > ### Author Response · Authors · 2025-08-05
> > > **Thank you for your review**
> > >
> > > Thank you for increasing the score. We appreciate your time and the feedback.

---

### Note · Authors · 2025-08-12

Dear Reviewers, AC, and SACs,

We greatly appreciate your effort and time in the review process.

We have presented the first work to suggest that reasoning models express their confidence more accurately, supported by extensive experiments and analysis. We believe that our work will make a strong and valuable contribution on advancing the trustworthiness of LLMs.

Overall, we believe the reviewers have received our work positively:

- All three reviewers acknowledged the significance of our topic.
- All three reviewers commended the design and comprehensiveness of our experiments and ablations.
- Reviewer PPaK highlighted the novelty and potential impact of our work.

While some concerns were raised, we believe our rebuttal has thoroughly clarified and addressed all points with well-supported explanations. We are confident that incorporating the reviewers’ feedback would mostly require only minor write-up adjustments to clarify misunderstandings and to add additional information.

We again thank you for your careful review of our work.

---

### Decision · Program_Chairs · 2025-09-17

**Decision:**

Accept (poster)

**Comment:**

This paper investigates the confidence calibration of reasoning versus non-reasoning large language models (LLMs). This work finds that reasoning models, especially those exhibiting “slow thinking” behaviors such as backtracking and exploring alternatives, tend to demonstrate superior calibration of verbalized confidence, even when accuracy is similar. The paper further shows that non-reasoning models can partially benefit from in-context reasoning demonstrations, providing a practical path to improving calibration.

The reviewers agree that the topic is important and timely, as calibration is central to building trustworthy LLMs. The experiments are comprehensive, covering multiple models and datasets, and include ablations that disentangle aspects of reasoning behavior contributing to calibration. The paper is clearly written, and the rebuttal usefully clarified the distinction between verbalized confidence (the paper’s focus) and logit-based confidence (common in prior work). This helps establish novelty. Additional experiments (e.g., SimpleQA) strengthened claims by addressing concerns about reasoning models underperforming on knowledge-dense benchmarks.

There are also some weaknesses identified during the review and discussion.
- Some reviewers noted that the initial framing risked overstating novelty, since prior work has studied CoT and confidence. However, the rebuttal clarified that this paper’s contribution is specifically on verbalized confidence calibration in reasoning models, which is distinct and less explored.
- Concerns remain about prompt sensitivity and the modest effect sizes on certain datasets. While the authors provided robustness checks and additional explanations (e.g., why gaps are smaller on high-accuracy datasets like TriviaQA), the issue of generalizing across prompting strategies and decoding methods could be explored further.
- One reviewer suggested exploring multiple modes of confidence expression (e.g., sample-based consistency) beyond verbalized confidence, which the authors acknowledge as important but out of scope for this work.

The reviews reflect a progression from initial skepticism (borderline reject) to greater confidence in the paper’s contributions after rebuttal and discussion. While not all concerns are fully resolved—particularly around robustness and generalization—the paper makes a substantive empirical contribution by demonstrating that reasoning models are better calibrated in verbalized confidence, and by identifying mechanisms (slow thinking behaviors) that explain this effect. This adds useful insight for both the reasoning model and calibration communities.